# In Vivo Predictive Dissolution and Biopharmaceutic-Based In Silico Model to Explain Bioequivalence Results of Valsartan, a Biopharmaceutics Classification System Class IV Drug

**DOI:** 10.3390/pharmaceutics16030390

**Published:** 2024-03-13

**Authors:** Isabel Gonzalez-Alvarez, Alejandro Ruiz-Picazo, Ruben Selles-Talavera, Andres Figueroa-Campos, Virginia Merino, Marival Bermejo, Marta Gonzalez-Alvarez

**Affiliations:** 1Department Engineering Pharmacy Section, Miguel Hernandez University, San Juan de Alicante, 03550 Alicante, Spain; isabel.gonzalez@umh.es (I.G.-A.); alejandro.ruizp@umh.es (A.R.-P.); ruben.selles@alu.umh.es (R.S.-T.); marta.gonzalez@umh.es (M.G.-A.); 2Departamento de Farmacia y Tecnología Farmacéutica y Parasitología, Facultad de Farmacia, Universitat de València, Av. Vicente Andrés Estellés s/n, 46100 Valencia, Spain; afigueroa@ispch.cl (A.F.-C.); virginia.merino@uv.es (V.M.); 3Instituto Interuniversitario de Investigación de Reconocimiento Molecular y Desarrollo Tecnológico (IDM), 46010 Valencia, Spain

**Keywords:** gastrointestinal simulator, in vitro dissolution, weak acid, dissolution modeling

## Abstract

The purpose of this study was to predict the in vivo bioequivalence (BE) outcome of valsartan (VALS, BCS class IV) from three oral-fixed combination products with hydrochlorothiazide (HCTZ, BCS class III) (*Co-Diovan^®^ Forte* as reference and two generic formulations in development) by conducting in vivo predictive dissolution with a gastrointestinal simulator (GIS) and a physiologically based biopharmaceutic model (PBBM). In the first BE study, the HCTZ failed, but the VALS 90% CI of Cmax and the AUC were within the acceptance limits, while, in the second BE study, the HCTZ 90% CI of Cmax and the AUC were within the acceptance limits, but the VALS failed. As both drugs belong to different BCS classes, their limiting factors for absorption are different. On the other hand, the gastrointestinal variables affected by the formulation excipients have a distinct impact on their in vivo exposures. Dissolution tests of the three products were performed in a GIS, and a PBBM was constructed for VALS by incorporating in the mathematical model of the in vitro–in vivo correlation (IVIVC) the gastrointestinal variables affected by the excipients, namely, VALS permeability and GI transit time. VALS permeability in presence of the formulation excipients was characterized using the in situ perfusion method in rats, and the impact of the excipients on the GI transit times was estimated from the HCTZ’s in vivo results. The model was able to fit the in vivo BE results with a good prediction error. This study contributes to the field by showing the usefulness of PBBM in establishing in vitro–in vivo relationships incorporating not only dissolution data but also other gastrointestinal critical variables that affect drug exposure in BCS class IV compounds.

## 1. Introduction

Generic drug development has provided a suitable pathway for the access to affordable, safe, efficacious, and high-quality medicines, allowing patients to fulfill their treatment needs and health systems to contain their healthcare expenditure.

The biopharmaceutics classification system (BCS) is a scientific framework allowing one to categorize drugs according to their aqueous solubility and intestinal permeability, together with the dosage form’s dissolution rate, all of which are the main factors which control drug absorption via the oral route. However, oral drug absorption may be altered by some excipient effects. This BCS system turned out to be a reference for generic drug development and has promoted alternative methods to in vivo studies to assess the BE of oral dosage forms, making it attractive from an ethical and economical point of view.

For BCS class I and class III drugs (highly soluble drugs), regulatory entities have established biowaivers as a way to approve new generics by performing comparative in vitro dissolution studies at pH buffers representing the range of values found in the GI tract in order to assure an equivalent performance in vivo [1,2]. For class II drugs (low solubility, high permeability), it is necessary to explore in vitro dissolution methods predictive of in vivo results and establish IVIVCs based on in vivo BE studies [3,4,5]. Once established and validated, post-approval changes can be supported based on in vitro dissolution studies exclusively.

In the past, it was assumed that excipients were inert and did not interfere with the drug absorption process. Currently, there is enough of a scientific basis to confirm that an excipient can modify absorption by altering a drug’s permeability and solubility as well as its intestinal transit time.

The purpose of physiologically based pharmacokinetic modeling (PBPK) for biopharmaceutical applications (PBBM) is to combine biopredictive dissolution/dissolution modeling and/or other in vitro testing parameters with PBPK modeling strategies to quantitatively describe (or characterize) the potential interactions of formulation variants with the organism and their effect on drug exposure. This modeling approach should include relevant mechanisms related to the absorption process, such as the local metabolism in the gastrointestinal tract (if applicable), drug transport properties, gastrointestinal transit, food state, etc., and incorporate a drug product’s quality properties to predict systemic drug exposure. This approach is the one contained in a recent FDA draft guidance that indicates how modeling can construct a safe space for variations in critical biopharmaceutical attributes [6,7]. This approach is not limited to the construction of IVIVCs, in which the in vivo dissolution process is the limiting step of absorption which can be captured in an in vitro dissolution test, both linked with a mathematical model. For BCS class III and some class IV drugs containing immediate-release (IR) products, establishing an IVIVC is challenging or unattainable, as permeability, instead of dissolution, is the limiting factor for systemic exposure.

VALS is a BCS class IV [8,9] weak acid drug with low solubility and permeability; thus, its systemic exposure depends on both factors, that is, permeability and its release from the dosage form. In two previous studies, two combination products of VALS and HCTZ (class III drug) failed to show BE. In the first study, HCTZ was not shown to be BE, but the 90% confidence interval (90%CI) VALS Cmax and the AUC were within the acceptance limits, while, in the second study, BE was shown for HCTZ, but not for VALS. As both drugs belong to different BCS classes, their limiting factors for absorption are different. In addition, the gastrointestinal variables affected by the formulation excipients have a distinct impact on their in vivo exposures. In this work, an in vivo predictive dissolution method in the GIS was used to characterize the dissolution behavior of both combination products. A PBBM was constructed for VALS by incorporating the gastrointestinal variables affected by the excipients, namely, VALS permeability and GI transit time, in the mathematical model of the IVIVC. VALS permeability in the presence of the formulations’ excipients was characterized with the in situ perfusion method in rats, and the impact of the excipients on the GI transit times was estimated from the HCTZ’s in vivo results.

## 2. Materials and Methods

### 2.1. Chemicals

VALS active pharmaceutical ingredient (API) was kindly provided by a pharmaceutical company. Acetonitrile, methanol, NaOH, NaCl, and NaH_2_PO_4_·H_2_O were purchased from Sigma-Aldrich (Barcelona, Spain). Purified water (i.e., filtrated and deionized) was used in the analytical methods and in vitro dissolution studies to prepare the dissolution media (Millipore, Billerica, MA, USA).

### 2.2. Drug Products

Two test products and one reference product were kindly provided by a pharmaceutical company. These products were IR tablets containing 320 mg of VALS and 25 mg of HCTZ. The reference product is commercialized in Europe as *Co-Diovan^®^ Forte* (Co-Diovan in all tables and figures). The test products were generic candidates that were tested in two different crossover BE studies in healthy subjects under fasting conditions. The outcome of both studies is shown in Table 1.

The first test product, designated as VALS_BE/HCTZ_Low, was bioequivalent for VALS C_max_ and AUC_0–t_ but failed for the C_max_ of HCTZ. The second test product, designated as VALS_Supra/HCTZ_BE, failed to show bioequivalence for both C_max_ and AUC_0–t_ but demonstrated BE for HCTZ. Both studies were single-dose, randomized, open-label, two-period, two-sequence, two-treatment, single-center, crossover pivotal BE studies with 48 subjects. The first one was completed by 44 subjects and the second by 46 subjects. In each study, the volunteers received two products: one dose of the IR test product (320 mg VALS/25 mg HCTZ) and one dose of the reference product (*Co-Diovan^®^ Forte* 320/25 mg). Blood samples were collected over a 48 h and 72 h interval in the first and the second study, respectively. The VALS and HCTZ concentrations in the plasma samples were determined using validated HPLC methods in both studies. The plasma C_max_ and AUC_0–t_ were calculated from the individual plasma concentration–time profiles by means of non-compartmental methods (linear trapezoidal rule).

The reference product did not contain SLS or sorbitol. Other non-critical excipients were similar across formulations. In the reference product, the excipients were as follows: cellulose microcrystalline, crospovidone, anhydrous colloidal silica, and magnesium stearate; and, In the film coating, hypromellose, macrogol 4000, talc, red iron oxide (E 172ii), black iron oxide (E 172i), and titanium dioxide (E 171). In the test products, the core excipients were the following: silicified microcrystalline cellulose, crospovidone, anhydrous colloidal silica, sorbitol, magnesium carbonate, pre-gelatinized starch, povidone, sodium stearyl fumarate, and SLS; and the film-coating was OPADRY OYL-28900 WHTIE, in a yellow color. Both test products contained the same amount of sodium lauryl sulphate (SLS), and the amount of sorbitol was reduced from 37 mg in VALS_BE/HCTZ_Low to 18.5 mg in VALS_Supra/HCTZ_BE.

### 2.3. Dissolution Experiments in GIS

The GIS is composed of three chambers, which represent the stomach, duodenum, and jejunum. After the drug product was administered into the gastric chamber, the gastric contents were pumped into the duodenal and subsequent jejunal chambers through a transfer tube (Figure 1).

A tablet of each VALS/HCTZ combination product was added to the stomach compartment at the start of the study. The dissolution media, initial volumes, and secretion rates are described in Table 2.

The transfer rate from the stomach to the duodenum was set as a first-order kinetic process with a gastric half-life of 8 min, in accordance with human data [11,12]. The volume of the stomach was decreasing from 300 mL to 10 mL. The stomach received acid secretions from a reservoir at a fixed rate (1 mL/min). The second chamber represented the duodenum, whose volume remained constant at 50 mL. The duodenum received the contents of the stomach and buffer solution at the desired pH at 1 mL/min. The output pump balanced its speed with the inputs so that the volume remained constant. The last chamber represented the jejunum. This compartment was empty at the beginning of the experiment, and it was the final accumulative receiver. To sum up, there were five compartments—stomach, duodenum, jejunum, and two reservoirs of secretion fluids—and all of them were connected by four pumps (Ismatec REGLO pump; IDEX Health and Science, Glattbrugg, Switzerland). All the peristaltic pumps were calibrated prior to the start of the experiment. All the compartments were in a thermostatic bath at 37 °C. The dissolution vessels were stirred with paddles (Muscle Corp., Osaka, Japan) at a rate that operated discontinuously—quickly for five seconds and slowly for 25 s. In such a setup, pH meters could also be inserted for continuous pH measuring.

Sampling at different times allowed the evaluation of the dissolution profile in each chamber. The samples were immediately centrifugated at a speed of 17,000× *g* (AccuSpin Micro 17, Fisher Scientific, Pittsburgh, PA, USA), and the supernatant was diluted (50:50) in the corresponding buffer.

All the components of the system (pumps and overhead paddles) were controlled by an in-house computer software program.

### 2.4. In Situ Permeability Determination in Rats

The VALS permeability values were experimentally obtained by the in situ closed-loop perfusion technique (Doluisio’s Method) [13]. The perfusion experiments were carried out using the whole small intestine of a rat.

Isolated segments in the complete small intestine (≈100 cm) were created. In order to remove all the intestinal contents, the intestine was copiously flushed with a physiologic isotonic solution (1% Sörensen phosphate buffer (*v*/*v*), 37 °C). When the surgical procedure was finished, the abdomen was covered with a cotton wool pad, avoiding peritoneal liquid evaporation and heat losses. The drug solution was introduced inside the compartment, and the samples were collected every 5 min up to a period of 30 min. In situ experiments were carried out with the three formulations (Co-Diovan, VALS_BE/HCTZ_Low, and VALS_Supra/HCTZ_BE) predissolved in a phosphate-buffered solution at a pH of 7.

Upon completion of the experiments, there was a decrease in the volume of the infused solutions as a result of water being reabsorbed. Consequently, it became necessary to make adjustments in order to accurately calculate the absorption rate constants. The process of water reabsorption was identified as a zero-order phenomenon. To determine the zero-order constant for water reabsorption (k_0_), a method was employed that involved directly measuring the remaining volume of the test solution. The initial volume of the experiment (V_0_) consisted of the volume of the drug solution (10 mL for the entire small intestine) plus the residual volume after flushing the intestinal segment. This residual volume had already been characterized and typically ranges from 0.3 to 0.5 mL. The final volume of the experiment (V_end_) was measured for each animal by carefully extracting and squeezing the intestinal segment. An individual value of ko was estimated for each animal using the following formula:(1)$$k_{0}=\left(V_{0}-V_{end}\right)/t_{end}$$

The measured volume at the end of the experiment (t_end_ = 30 min) in each animal is denoted as V_end_ in the formula above. The k_0_ value is utilized to estimate the remaining water volume in the various segments at each time point (V_t_). To obtain the actual C_t_, the experimentally analyzed drug concentrations (C_e_) were adjusted at each time point using the following equation:(2)$$C_{t}=C_{e}\left(V_{t}/V_{0}\right)$$

In the absence of water reabsorption, the drug concentration in the gut at a specific time is denoted as C_t_. The experimental value is represented by C_e_. The corrected concentrations (C_t_ values) are utilized to determine the absorption rate coefficients [14].

A nonlinear regression analysis was used to determine the absorption rate coefficient (k_a_) by comparing the remaining concentrations in the lumen (C_t_) over time.
(3)$$C_{t}=C_{0}\times e^{\left({-k}_{a}\times t\right)}$$

The permeability value was derived from the given k_a_ value using the following correlation:(4)$$P_{eff}=k_{a}\times \frac{R}{2}$$

The effective radius of the intestinal segment, denoted as R, can be determined by considering the segment as a cylindrical shape. The calculation of R takes into account the relationship between the dimensions of the segment and its cylindrical representation:(5)$$Volume=\pi \times R^{2}\times L$$

To estimate the parameters, a perfusion volume of 10 mL was employed for the entire small intestine. The length of the small intestine (L) was determined to be 100 cm in its entirety.

Male Wistar rats weighting 250–300 g (n = 4–6) were used for all the in situ permeability studies. The animal experiments were approved by the Scientific Committee of the Faculty of Pharmacy, Miguel Hernandez University, and followed the guidelines described in the EC Directive 86/609, the Council of the Europe Convention ETS 123, and Spanish national laws governing the use of animals in research.

The in situ experiments were carried out with the three formulations (Co-Diovan, VALS_BE/HCTZ_Low, and VALS_Supra/HCTZ_BE) predissolved in a phosphate-buffered solution at a pH of 7.

### 2.5. HPLC Analytical Method

Samples from the experimental assays were analyzed by HPLC using a UV detector (Waters^®^ 2487) and an X-Bridge^®^ C18 column (3.5 μm, 4.6 × 100 mm). The mobile phase was a mixture of 50% methanol and 50% acid water (0.05% *v*/*v* TFA in water), with a flow of 1 mL/min. The peaks were observed after 2.5 min, and the wavelength was set to 225 nm.

### 2.6. Analysis of the Mass Transport of VALS throughout the GIS

A mass transport analysis approach was used to describe the evolution with time of the VALS concentrations in all the GIS chambers. The mathematical model was based on differential equations including the dissolution, precipitation, and transit kinetics. The model was based on the one previously described by Matsui and colleagues [15]. In order to account for the VALS properties, the model was modified (Table 3), and, thus, the precipitation process was incorporated only in the stomach. Equations are described in the Appendix A.

The model was coded in Phoenix WinNonlin V8 (Certara USA, Princeton, NJ, USA), and the fitting to the experimental data was performed with a simplex algorithm.

### 2.7. In Silico Model to Predict the Pharmacokinetic (PK) Profiles of VALS

The disposition kinetics of VALS after the administration of the VALS/HCTZ combination products was described with a two-compartmental PK model. The model was based on the one previously described by Matsui et al. [15] but incorporated modifications to accommodate the weak acid characteristics of VALS (Figure 2).

This model represents a central and peripheral compartment for VALS disposition, a precipitation process in the stomach, and a pH-dependent dissolution process. Useful PK parameters to include in the model were extracted from the fitting of the reference product’s plasma levels to a two-compartment pharmacokinetic model with first-order absorption. The absorption rate constant (k_a_) value was experimentally determined in rats by the in situ closed-loop perfusion method. The k_a_ values were transformed into the permeability ones as P_eff_ = k_a_ × R/2 with 0.18 cm as the R value in the rats. The rat permeability values were scaled up to the human values with an average conversion factor of 4 based on previously published human–rat correlations [16]. The P_eff_ was assumed to be the same in the duodenum and jejunum segments due to the fact that no experimental permeability in the intestinal segmental has been published.

## 3. Results

### 3.1. Performance of VALS from the VALS/HCTZ Combination Products in the GIS

Figure 3 depicts the average VALS dissolved amounts (four tablets per formulation) in the three chambers from the assayed products. The fitted values to the PBPK model were represented with dashed lines.

The precipitation and dissolution curve-fitted coefficients from the in vitro GIS model are displayed in Table 4.

### 3.2. In Silico PBBM to Forecast the Systemic Performance of VALS in the VALS/HCTZ Combination Products

The plasma profiles of the VALS products were used to fit a PBBM in which the in vitro dissolution data in the GIS were used as the input and convoluted with the pharmacokinetic two-compartment disposition parameters obtained from the reference formulation. It was a numerical convolution by means of the integration of the differential equations. The PK parameters are listed in Table 5.

The experimental permeability values in rats (calculated from the absorption rate coefficients) scaled up to human values were used for each formulation (Table 6).

The final permeability values are displayed in Table 7.

The gastric emptying times were shortened (from 15 to 13 min) for the VALS_Supra/HCTZ BE formulation. These gastric emptying times were based on previous human intubation studies (13 min) with a 2 min delay in the formulation containing more sorbitol, based on the AUC and C_max_ ratio differences for the HCTZ in both test products. The total transit time in the absorption window (quantified with tcut) was reduced from 600 min to 360 min for VALS_BE/HCTZ_Low. The precipitation and dissolution parameters from the in vitro model were used as the initial estimates in the PBBM, and the final parameters were compared with the initial ones to estimate an average common scaling factor between the in vitro and in vivo parameters. The scaling factors and the final in vivo dissolution/precipitation parameters are displayed in Table 7. These parameters were used to simulate the in vivo plasma concentration–time profiles using as input the in vitro dissolution profiles in the GIS. The model equations are detailed in the Appendix A. The experimental and simulated plasma profiles are illustrated in Figure 4.

The predicted C_max_ and AUC_0–t_ values were calculated using non-compartmental methods (linear trapezoidal rule). Table 8 shows the prediction errors of the C_max_ and AUC_0–t_ values.

## 4. Discussion

The excipient impact on VALS dissolution/precipitation in the stomach was reflected in the GIS apparatus. The original differences in stomach were maintained in the next chamber (duodenum) and also kept in the jejunal chamber (Figure 3). This means that the concentrations reaching the absorbing sites will be different in the three products, leading to different systemic exposures (Table 1 and Table 8). The in vitro dissolution results along with the in silico simulations indicated that one generic product would be similar to the reference product for VALS, whereas the other generic product would not be similar for VALS. These predictions agreed with the systemic exposure data obtained from crossover BE studies. Consequently, this dissolution device and media conditions could be useful for the internal decision-making process of generic pharmaceutical companies when developing their formulations.

VALS, as a weakly acidic compound, has a higher solubility in the intestinal environment, and its dissolution in stomach is incomplete. Interestingly, the difference among the three oral tablets become evident already in the gastric chamber (Figure 3). The different behavior of both test products could be explained due to their different sorbitol content. It has been reported that highly soluble and/or hygroscopic ingredients, like sorbitol, decrease the effectiveness of superdisintegrants like crospovidone [17]. The in vitro disintegration times for the three formulations were similarly fast (less than 5 min), but the pharmacopoeial disintegration conditions might not reflect the in vivo ones.

Sorbitol’s osmotic effect in altering gastric emptying and intestinal motility has been previously reported but, generally, in the presence of higher sorbitol amounts than those in the formulations used in this study [18,19,20], although the effect of a few milligrams of sorbitol has also been described to affect the absorption of risperidone [18]. From these results, it seems feasible to accept the hypothesis that even small amounts of this excipient can affect the overall intestinal transit of a drug, which justifies the reduction in the overall intestinal transit time in our study (tcut), and, due to the different biopharmaceutical properties of VALS and HCTZ, even a small change in gastric emptying and transit time through the absorption window could be reflected in a lower C_max_ for HCTZ without reducing the C_max_ and AUC of VALS. Therefore, in our study, the higher sorbitol content of VALS_BE/HCTZ_Low (37 vs. 18.5 mg) was reflected in a slightly slower gastric emptying process (15 min as GE t_1/2_) but a shorter transit time in the intestine [19,20,21], reducing the residence time in the absorption window for HCTZ (from 600 min to 360 min). The formulation with less sorbitol showed no apparent effect on the gastric emptying time (13 min as GE t_1/2_). CoDiovan Forte, on the other hand, could be correctly predicted with a GE t_1/2_ of 13 min and 600 min as its overall residence time in the absorption window, like VALS_Supra/HCTZ_BE. Absorption does not cease after Cmax. This point represents the moment at which the input and output rates from the body are equal, and, after this moment, absorption will continue; meanwhile, dissolved drug can be found during the absorption window.

A total of 360 min was selected as the transit time for the formulation containing the highest sorbitol amount (so a shorter residence in the intestine), considering the mean residence time in both segments and taking a slightly shorter value. Then, for the formulations not containing sorbitol, a long-enough value was selected due to the fact that it does not limit absorption.

The precipitation and dissolution parameters from the in vitro model were used as the initial estimates in the PBBM, and the final parameters were compared with the initial ones to estimate an average common scaling factor between the in vitro and in vivo parameters.

In addition, the presence of sodium lauryl sulphate in both test formulations resulted in higher permeability values for VALS in rats for both test products, which, when incorporated in the PBBM model, gave predictions which were consistent with the in vivo outcomes. However, it is not known why one of the formulations exhibited a larger P_eff_ value if both contained the same amount of SLS. Quantitative differences in other excipients must have been affecting the P_eff_ value or the activity of SLS. The advantage of determining the in situ permeability of the whole excipient mixture in the formulations in the rat intestines was that the complex interplay between the excipients was then reflected in the obtained P_eff_ value.

In summary, the origin of the failure of VALS seemed to be a more efficient disintegration/dissolution in the stomach coupled with a higher VALS permeability, while, for HCTZ, failure was associated with the differences in the gastric emptying rate and the shorter residence time in the absorption window. The absence of osmotic sorbitol’s effect on VALS compared to HCTZ could have been due to a higher permeability of VALS (41.6 × 10^−4^ cm/s as the lowest value reported in this study for VALS versus 0.26 × 10^−4^ cm/s for HCTZ in rat jejunum, reported by Liu Z. et al. [22].

The objective of model fitting the in vitro data was to obtain initial estimates of the parameters to scale up for the in vivo model. There are clear discrepancies between the observed and model-predicted data in Figure 3, but, as the rank order of the fitted lines was correct and the evolution of the amounts with time had been reproduced, it was not considered necessary to make the model more complex. On the other hand, when a more complex model was used with precipitation in all the compartments, the uncertainty in the estimated values was very high, and a simpler model with the same dissolution coefficient in all the compartments did not reproduce the experimental data. The simulated profiles in Figure 4 also show discrepancies with the experimental data, but, as the key pharmacokinetic parameters had been well predicted, introducing more parameters or processes in the model was not considered essential.

The plasma concentration predictions were obtained with a deterministic model without including parameter variability or residual variability, so the 90% confidence intervals of the predictions cannot be calculated. Nevertheless, if used in the generic development setting to inform researchers of the risk of bioequivalence failure, the proposed in vitro dissolution methodology combined with the permeability experiments and the change in the intestinal transit time would have identified a “non-BE” VALS/HCTZ combination product for VALS.

## 5. Conclusions

As VALS and HCTZ belong to different BCS classes, the gastrointestinal variables affected by the excipient’s composition have a distinct impact on their in vivo exposures. Whereas, in our study, the accelerated intestinal transit time affected the C_max_ of HCTZ, the PK of VALS was not affected.

The in vitro dissolution tests of the three combination products in a GIS apparatus showed the different behaviors of the formulations, which, in part, explained the BE failure for VALS in one of the test products.

The in vitro dissolution results combined with a PBBM incorporating the gastrointestinal parameters affected by the excipients, namely, VALS permeability and the GI transit time, were able to predict the plasma concentrations–time profiles of VALS obtained from the BE studies with good prediction errors.

This study contributes to the field by showing the usefulness of PBBM modeling in establishing in vitro–in vivo relationships incorporating not only in vitro dissolution data but also other critical gastrointestinal variables that affect drug exposure in BCS class IV compounds.

## Figures and Tables

**Figure 1 pharmaceutics-16-00390-f001:**
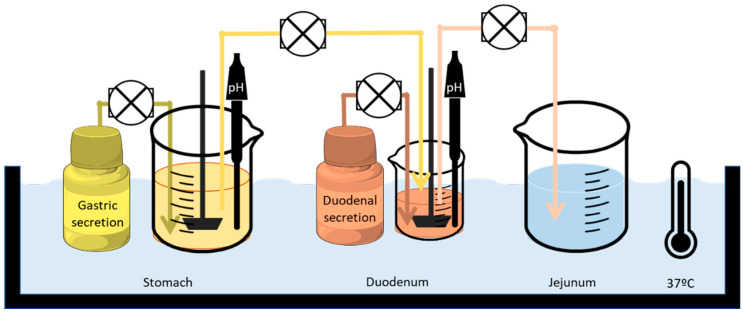
Setup and design of the GIS that was applied to test the different products of VALS in fasted-state conditions. Adapted from [10] with permission.

**Figure 2 pharmaceutics-16-00390-f002:**
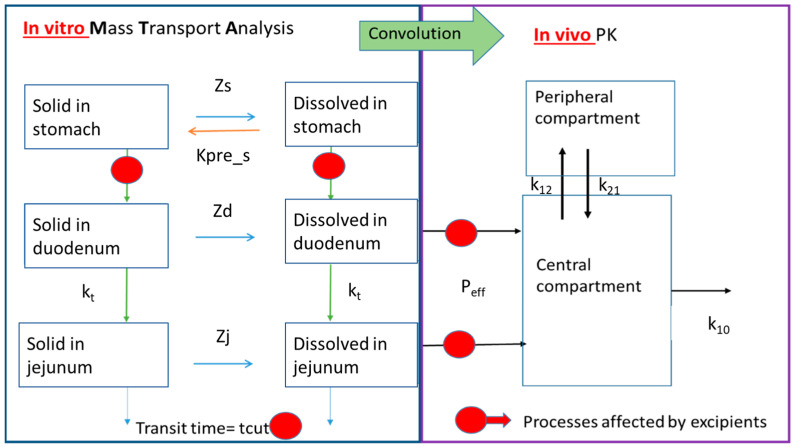
Mass transport analysis model (MTA) of the in vitro data as observed in the GIS with the VALS/HCTZ combination products coupled with the in silico PK parameters in order to predict the plasma profiles of VALS. The gastric emptying times, permeability values, and total transit time (tcut) are changed for each formulation to account for the excipient effects on these parameters. Z (segment): dissolution rate coefficient in s—stomach, d—duodenum, and j—jejunum; kpre_s: precipitation rate constant in the stomach; k_t_: transit constant from the jejunum to the distal segments; P_eff_: permeability value; and k_10,_ k_12_, and k_21_: distribution constants.

**Figure 3 pharmaceutics-16-00390-f003:**
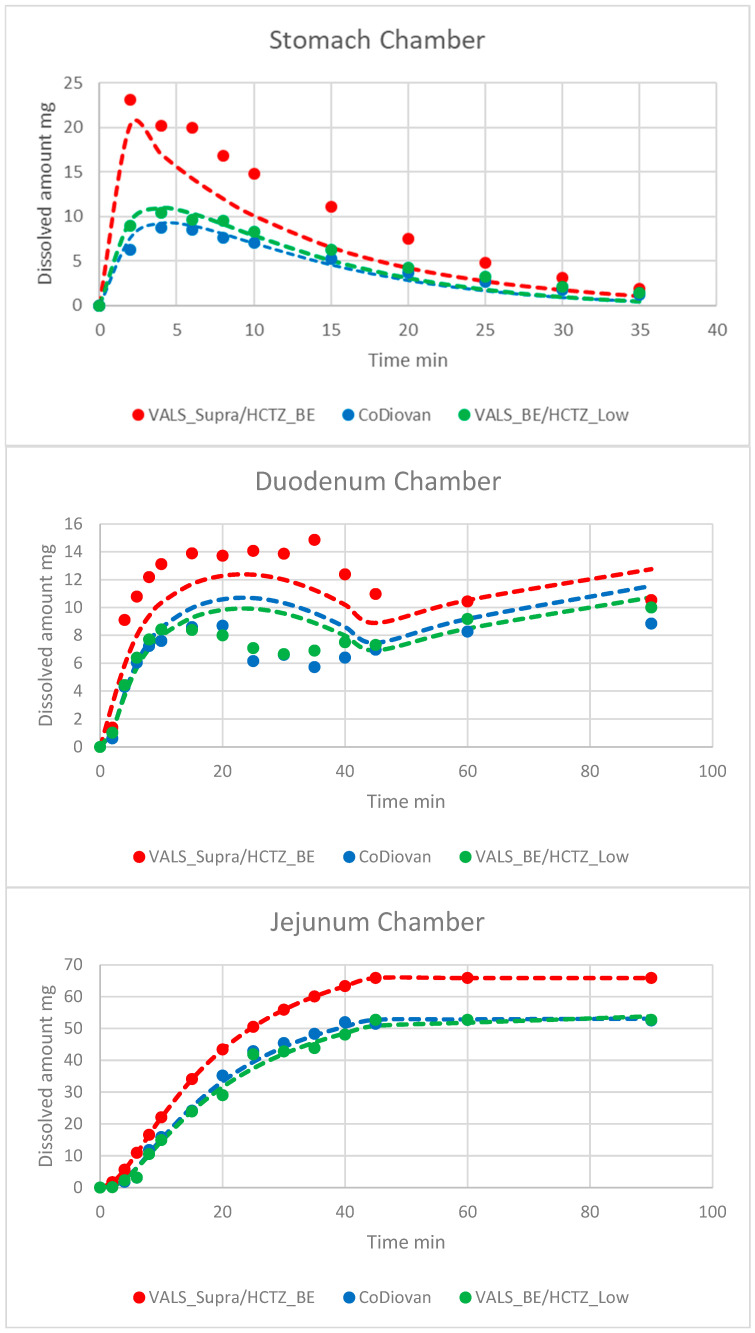
Experimental (dots) and model-predicted (dashed lines) values of VALS amounts dissolved for reference product CoDiovan Forte and the test products in each GIS chamber.

**Figure 4 pharmaceutics-16-00390-f004:**
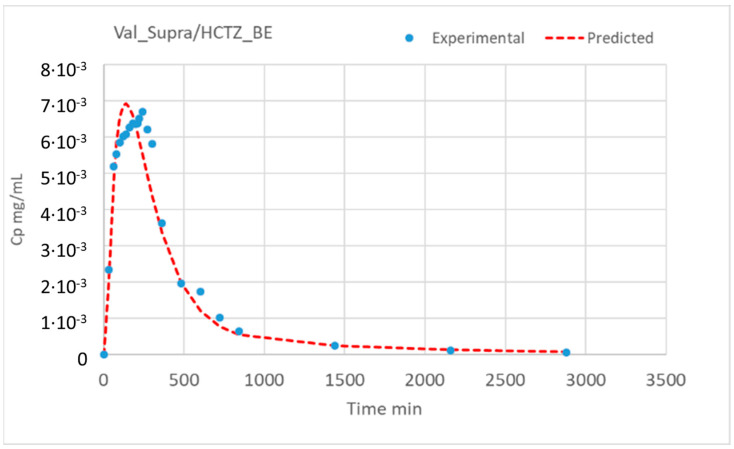
Experimental (dots) and simulated plasma concentration–time profiles (dashed lines) of VALs in the reference and test VALS/HCTZ combination products.

**Table 1 pharmaceutics-16-00390-t001:** 90% CIs in both in vivo BE studies.

90% CI		First Study VALS_BE/HCTZ_Low	Second Study VALS_Supra/HCTZ_BE
HCTZ	AUC_0–t_	88.48–96.03	94.84–104.15
	C_max_	**78.28–87.48**	84.74–96.28
VALS	AUC_0–t_	91.42–113.27	**102.88–127.06**
	C_max_	87.74–109.50	**107.61–140.35**

**Table 2 pharmaceutics-16-00390-t002:** Experimental conditions in the GIS for testing the different drug products of VALS.

Fasted-State Test Conditions	GIS Stomach	GIS Duodenum	GIS Jejunum
Dissolution Media	Simulated gastric fluid (SGF), pH 2.0, 0.01 M HCl + 34.2 mM NaCL	Phosphate buffer 50 mM pH 6.8	/
Initial Volume	50 mL SGF + 250 mL of water	50 mL	/
Secretions	1 mL/min of SGF	1 mL/min of Phosphate buffer 100 mM pH 6.8	/

**Table 3 pharmaceutics-16-00390-t003:** Input parameters for describing the dissolution, precipitation, and transit kinetics of VALS for three different oral products in the GIS.

	VALS_Supra/HCTZ_BE	VALS_BE/HCTZ_Low	CoDiovan	Reference
Dose (mg)	320/25	320/25	320/25	
k_sec_s_ (mL/min)	1	1	1	[12]
k_sec_d_ (mL/min)	1	1	1	[12]
t_1/2 G_ (min)	8	8	8	[12]
V_s_ (mL)	300 to 5	300 to 5	300 to 5	[12]
V_d_ (mL)	50	50	50	[12]
V_j_ (mL)	0 to 390	0 to 390	0 to 390	[12]
Cs at pH 2.0 (mg/mL)	0.08	0.08	0.08	[9]
Cs at pH 4.5 (mg/mL)	1.23	1.23	1.23	[9]
Cs at pH 6.8 (mg/mL)	9.21	9.21	9.21	[9]

k_sec_s_ and k_sec_d_: secretion rates in stomach and duodenum, respectively; t_1/2G_: gastric emptying half-life; V_s_, V_d_, and V_j_: volumes in stomach, duodenum, and jejunum; and Cs: solubility values at each pH.

**Table 4 pharmaceutics-16-00390-t004:** Precipitation and dissolution curve-fitted coefficients from the in vitro GIS model.

	VALS_Supra/HCTZ_BE	VALS_BE/HCTZ_Low	CoDiovan
k_pre_s_ (min^−1^)	5.92 × 10^−7^	2.41 × 10^−3^	1.99 × 10^−3^
Z_s_ (mL/mg/min)	2.87 × 10^−1^	3.26 × 10^−5^	2.42 × 10^−5^
Z_d_ (mL/mg/min)	2.87 × 10^−4^	1.94 × 10^−4^	2.35 × 10^−4^
Z_j_ (mL/mg/min)	2.30 × 10^−13^	3.75 × 10^−9^	4.20 × 10^−10^

Z_(segment)_: dissolution rate coefficient in s—stomach, d—duodenum, and j—jejunum; and k_pre_s_: precipitation rate constant in the stomach. The in vitro fitted parameters were used as the initial estimates for the PBBM with a one-step convolution IVIVC.

**Table 5 pharmaceutics-16-00390-t005:** Pharmacokinetic parameters of VALS.

Parameter (Units)	Value
Vc/Fsys (mL)	25,120.7
k_10_ (h^−1^)	0.26753
k_12_ (h^−1^)	0.06743
k_21_ (h^−1^)	0.05925

Vc is the distribution volume. Fsys is the systemic availability that incorporates the oral fraction absorbed and the first-pass effect of Valsartan. k_10_ is the first-order elimination rate constant from the central compartment, and k_12_ and k_21_ are the distribution coefficients from the central to the peripheral and the peripheral to the central compartments, respectively.

**Table 6 pharmaceutics-16-00390-t006:** Absorption rate constant (k_a_) of the three formulations.

Parameter (Units)	k_a_ (h^−1^)
CoDiovan	0.74
VALS_Supra/HCTZ_BE	0.89
VALS_BE/HCTZ_Low	1.01

**Table 7 pharmaceutics-16-00390-t007:** Final parameters of the PBBM for VALS.

	VALS_Supra/HCTZ_BE	VALS_BE/HCTZ_Low	CoDiovan	Reference
R (cm)	1.5	1.5	1.5	[11]
P_eff_ (cm/min)	59.5 × 10^−4^	47.0 × 10^−4^	41.6 × 10^−4^	Experimental values in rat scaled to human values
t_1/2 G_ (min)	13	15	13	[11]
Fsys	0.39	0.39	0.39	[11]
k_t_ (min^−1^)	0.0056	0.0056	0.0056	[15]
tcut (min)	600	360	600	[11]
k_pre_s_ (min^−1^)	1.18 × 10^−7^	4.82 × 10^−4^	3.98 × 10^−4^	Scaled from the in vitro parameter (×0.2)
Z_s_ (mL/mg/min)	1.15	1.30 × 10^−4^	9.68× 10^−5^	Scaled from the in vitro parameter (×4)
Z_d_ (mL/mg/min)	1.15 × 10^−3^	7.76 × 10^−4^	9.40 × 10^−4^	Scaled from the in vitro parameter (×4)
Z_j_ (mL/mg/min)	2.30 × 10^−12^	3.75 × 10^−8^	4.20 × 10^−9^	Scaled from the in vitro parameter (×10)

R: intestinal radius; P_eff_: effective permeability; t_1/2G_: gastric emptying half-life; Fsys: systemic availability; k_t_: transit constant from jejunum to distal segments; tcut: total transit time; k_pre_s_: precipitation rate constant in stomach; and Z_(segment)_: dissolution rate coefficient in s—stomach, d—duodenum, and j—jejunum.

**Table 8 pharmaceutics-16-00390-t008:** Prediction errors of the PBBM for the PK parameters of VALS.

**C_max_ (mg/mL)**	**Experimental**	**Predicted**	**Error (%)**
CoDiovan Forte	5.73 × 10^−3^	5.75 × 10^−3^	0.37
VALS_BE/HCTZ_Low	5.93 × 10^−3^	6.39 × 10^−3^	7.67
VALS_Supra/HCTZ_BE	6.70 × 10^−3^	6.94 × 10^−3^	3.44
**AUC_0–t_ (mg/mL·min)**	**Experimental**	**Predicted**	**Error (%)**
CoDiovan Forte	2.93	2.90	−1.32
VALS_BE/HCTZ_Low	3.03	3.14	3.75
VALS_Supra/HCTZ_BE	3.17	2.95	−6.96

## Data Availability

Data are contained within the article.

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
