# Peer review of "In Vivo Predictive Dissolution and Biopharmaceutic-Based In Silico Model to Explain Bioequivalence Results of Valsartan, a Biopharmaceutics Classification System Class IV Drug"

_pharmaceutics, 2024, doi:10.3390/pharmaceutics16030390_

Round 1

Reviewer 1 Report

Comments and Suggestions for Authors

Current studies focus on predicting the bioequivalence outcome of valsartan from three oral fixed combination products with hydrochlorothiazide using in vivo predictive dissolution and a physiologically-based biopharmaceutic model. 

Accept after minor revision.

1. Some minor errors in the sentence need to be addressed such as line number 79, "VALS is a BCS class IV weak acid with low solubility and permeability, 79 thus, its systemic exposure depends on both factors, permeability and release 80 from the dosage form" write weak acid "drug "

2. Add supportive literature that supports that Valsartan is a BSC class IV drug. Some literature suggests that Valsartan is a BCS class II and some are suggesting a BCS class III drug (Ref. https://doi.org/10.1208/s12249-018-1028-x and  10.2174/2589977511666191210151120). 

3. Figure 2 needs more clarity related to mass transport. The image indicated that the stomach was solid or dissolved. is it possible to modify terminology such as solid in stomach or drug (specific name if possible) on stomach? 

Author Response

Reviewer 1.

Thanks for your comments. Your comments have contributed to improve our paper. Thansk so much

  1. Some minor errors in the sentence need to be addressed such as line number 79, "VALS is a BCS class IV weak acid with low solubility and permeability, 79 thus, its systemic exposure depends on both factors, permeability and release 80 from the dosage form" write weak acid "drug "

The word “drug” has been added after weak acid

  1. Add supportive literature that supports that Valsartan is a BSC class IV drug. Some literature suggests that Valsartan is a BCS class II and some are suggesting a BCS class III drug (Ref. https://doi.org/10.1208/s12249-018-1028-x and  2174/2589977511666191210151120). 

Low solubility is reported in reference 8  (Hamed & Alnadi, 2018) which has been included. Low permeability is reported in reference 9 (de Castro et al.) (as reviewer suggested) based on experimental in vitro values and human fa values and in reference 10 (Lozoya et al) based on rat perfusion studies. The three references have been added after the statement that Valsartan is a BCS Class IV drug in line 79.

  1. Figure 2 needs more clarity related to mass transport. The image indicated that the stomach was solid or dissolved. is it possible to modify terminology such as solid in stomach or drug (specific name if possible) on stomach? 

Figure 2 has been modified accordingly to reviewer suggestion. Your can see in the modify version (attached)

Reviewer 2 Report

Comments and Suggestions for Authors

The manuscript reports in vivo predictive dissolution with the GastroIntestinal Simulator combined with in situ permeability studies and physiologically-based biopharmaceutic modeling to analyze dissolution, absorption and bioequivalence of valsartan-hydrochlorothiazide oral combination drug products. The manuscript presents interesting experimental approach, based on the data obtained in different experimental systems (in vitro and in vivo), and their modeling-based analysis. I have several comments in relation to the contents of this manuscript.

Major comments.

1. The description and graphical presentation of the in vivo model are not clear and should be revised.

Figure 2 – please state the names of the specific rate constants, using the same parameters as listed in Tables 4-7. The names, units, and the meaning of the parameters should be clear from the Figure 2 and the Methods, without the need to go to the Supplementary Materials.

Peff was assumed to be equal in the duodenum & jejunum, why? How is Fsys is defined? What is the transit time between the duodenum and jejunum? Why the tcut walue was set to 600 and 360 min (please compare to 4*t1/2 absorption of 165-225 min based on data from Table 6, and predicted Cmax of 100-200 min in Fig. 4). In general, the reasons for choosing the specific values for the parameters and scaling factors in Table 7 are not clear and should be explained in the manuscript.

2. There are substantial discrepancies between the observed vs. model-predicted data – in Fig. 3 and Fig. 4. Please state this explicitly in the Results and Discussion, describe the discrepancies, which models were considered and ruled out for analysis of the experimental data, etc.

3. Table 6 – the data of the in situ permeability studies for the 3 studied formulations and the relevant controls were not presented. Please add these data, and more detailed description of these experiments (perfusion site – duodenum, jejunum, all the small intestine; concentrations of valsartan/hydrochlorothiazide, controls, …). 

4. Why hydrochlorothiazide was not quantified and modeled? Lines 345-346 – potential reasons for lack of hydrochlorothiazide bioequivalence – is speculation, that does not match the results in Fig. 3 (differences between the formulations in the different chambers). Please consider to analyze the hydrochlorothiazide data and to present the results of this analysis.

Small corrections:

1. The terms in the title are non-standard and also appear to be misleading, especially given the discrepancies between the observed and predicted data (see above). The data actually originated from in vitro model and in vivo experiments (in situ perfusion and PK data from BEQ studies), and they were analyzed using a sort of minimal PBPK. Please revise and correct.

2. Line 17 – “hydrochlorothiazide” – please add the missing h in thiazide.

3. Line 136 – Opadry type and color – please check for typo errors.

4. Line 282 – “radius”, please correct.

Comments on the Quality of English Language

The English is fine. Only small typo corrections are needed (see the "Small corrections").

Author Response

Thanks for your comments. Your expertise has contributed to improve our paper

all comments have included in final version (attached)

Reviewer 2

  1. The description and graphical presentation of the in vivomodel are not clear and should be revised.

Figure 2 – please state the names of the specific rate constants, using the same parameters as listed in Tables 4-7. The names, units, and the meaning of the parameters should be clear from the Figure 2 and the Methods, without the need to go to the Supplementary Materials.

Figure 2 has been modified according to the reviewer suggestion, so the parameters can be clearly identified.

  1. Peff was assumed to be equal in the duodenum & jejunum, why?

Peff was assumed to be the same in both segments as we have no experimental data in each segment but only the in situ perfusion estimation in the whole small intestine of the rat.

This statement has been included in the manuscript.

Rat permeabilities were scaled up to human values with an average conversion factor of 4 based on previously published human-rat correlations (PMID: 17727800 and https://doi.org/10.1016/j.ejps.2017.06.033) (Table 5).) Peff was assumed to be the same in duodenum and jejunum segments due to no experimental permeability in intestinal segmental has been published

  1. How is Fsys is defined?

Fsys is the systemic availability that incorporate the oral fraction absorbed and the first pass effect of Valsartan. The definition of the term has been includen in the manuscript.

Fsys is the systemic availability that incorporate the oral fraction absorbed and the first pass effect of Valsartan. k10 is the first order elimination rate constant from central compartment and k12 and k21 are the distribution coefficients from central to peripheral and peripheral to central, respectively

  1. What is the transit time between the duodenum and jejunum?

VALS_Supra/HCTZ_BE

VALS_BE/HCTZ_Low

Co-Diovan

Reference

R (cm)

1.5

1.5

1.5

[9]

Peff (cm/min)

59.5 · 10-4

47.0 · 10-4

41.6 · 10-4

Experimental values in rat scaled to human

t1/2 G (min)

13

15

13

[9]

Fsys

0.39

0.39

0.39

[9]

kt (min-1)

0.0056

0.0056

0.0056

[14]

tcut (min)

600

360

600

[9]

kpre_s (min-1)

1.18 · 10-7

4.82 · 10-4

3.98 · 10-4

Scaled from in vitro parameter (x0.2)

Zs (mL/mg/min)

1.15

1.30 · 10-4

9.68· 10-5

Scaled from in vitro parameter (x4)

Zd (mL/mg/min)

1.15 · 10-3

7.76 · 10-4

9.40 · 10-4

Scaled from in vitro parameter(x4)

Zj (mL/mg/min)

2.30 · 10-12

3.75 · 10-8

4.20· 10-9

Scaled from in vitro parameter (x10)

The transit time rate constant between compartments was taken from previous references (indicated in the table 7).

Table 7. Final parameters of the PBBM for VALS.

R: intestinal radious; Peff: effective permeability; t 1/2G : gastric emptying half-life; Fsys: systemic availability; kt: transit constant from jejunum to distal segments; tcut: total transit time kpre_s: precipitation rate constant in stomach; Z(segment): dissolution rate coefficient in s: stomach; d: duodenum; j: jejunum.

  1. Why the tcut value was set to 600 and 360 min (please compare to 4*t1/2 absorption of 165-225 min based on data from Table 6, and predicted Cmax of 100-200 min in Fig. 4).

Absorption do not cease after Cmax. That point represents the moment in which input and output rates from the body are equal, and after that moment, absorption will continue meanwhile there is dissolved drug in the absorption window.

We selected 360 min as transit time for the formulation containing the highest sorbitol amount (so shorter residence in intestine) considering the mean residence time in both segments and taking a slightly shorter value. Then for the formulations not containing sorbitol we set an enough long value that does not limit absorption.

That information has been included in the manuscript

The formulation with less sorbitol showed no apparent effect on the gastric emptying time (13 min as GE t1/2). Co-Diovan Forte, on the other side, can be correctly predicted with a GE t1/2 of 13 min and 600 min as overall residence time on the absorption window like VALS_Supra/HCTZ_BE.)

Absorption do not cease after Cmax. That point represents the moment in which input and output rates from the body are equal, and after that moment, absorption will continue meanwhile there is dissolved drug in the absorption window.

360 min was selected as transit time for the formulation containing the highest sorbitol amount (so shorter residence in intestine) considering the mean residence time in both segments and taking a slightly shorter value. Then for the formulations not containing sorbitol, an enough long value was selected due to that does not limit absorption.

The precipitation and dissolution parameters from the in vitro model were used as initial estimates in the PBBM and the final parameters were compared with the initial ones to estimate an average common scaling factor between in vitro and in vivo parameters.

  1. In general, the reasons for choosing the specific values for the parameters and scaling factors in Table 7 are not clear and should be explained in the manuscript.

This paragraph has been included in the discussion

The formulation with less sorbitol showed no apparent effect on the gastric emptying time (13 min as GE t1/2). Co-Diovan Forte, on the other side, can be correctly predicted with a GE t1/2 of 13 min and 600 min as overall residence time on the absorption window like VALS_Supra/HCTZ_BE.)

Absorption do not cease after Cmax. That point represents the moment in which input and output rates from the body are equal, and after that moment, absorption will continue meanwhile there is dissolved drug in the absorption window.

360 min was selected as transit time for the formulation containing the highest sorbitol amount (so shorter residence in intestine) considering the mean residence time in both segments and taking a slightly shorter value. Then for the formulations not containing sorbitol, an enough long value was set due to that does not limit absorption.

The precipitation and dissolution parameters from the in vitro model were used as initial estimates in the PBBM and the final parameters were compared with the initial ones to estimate an average common scaling factor between in vitro and in vivo parameters.

  1. There are substantial discrepancies between the observed vs. model-predicted data – in Fig. 3 and Fig. 4. Please state this explicitly in the Results and Discussion, describe the discrepancies, which models were considered and ruled out for analysis of the experimental data, etc.

The following paragraph has been included in the discussion

The objective of model fitting of the in vitro data was to obtain initial estimates of the parameters to scale up for the in vivo model. There are clear discrepancies between the observed and model predicted data in Figure 3 but as the rank order of the fitted lines was the correct and the evolution of the amounts with time was reproduced, it was not considered necessary to make more complex the model. On the other hand, when a more complex model was used with precipitation in all the compartments, the uncertainty on the estimated values was very high and a simpler model with the same dissolution coefficient in all the compartents did not reproduce the experimental data. The simulated profiles in figure 4 showed also discrepancies with the experimental data but as the key pharmacokinetic parameters were well predicted, introducing more parameters or processes in the model was not considered essential.

  1. Table 6 – the data of the in situpermeability studies for the 3 studied formulations and the relevant controls were not presented. Please add these data, and more detailed description of these experiments (perfusion site – duodenum, jejunum, all the small intestine; concentrations of valsartan/hydrochlorothiazide, controls, …). 

A summary of the next explanation has been included in the material and method section:

VALS permeability values were experimentally obtained by in situ closed loop perfusion technique (Doluisio’s Method) [11,12]. Perfusion experiments were done using the whole small intestine of the rat.

The absorption rate coefficients and the permeability values of VALS formulations were determined in the whole small intestine (n= 6-7) using in situ “closed loop” perfusion method based in Doluisio Technique (Doluisio et al., 1969, Lozoya-Agullo et al., 2015a, 2016a; Lozoya-Agullo et al., 2016b). Briefly, male Wistar rats (body weight, 250-300 g) were anesthetized using a mixture of pentobarbital (40 mg/kg) and butorphanol (0.5 mg/kg). Isolated segments in the  complete small intestine (≈100 cm) were created. In order to remove all the intestinal contents the intestine was copiously flushed with a physiologic isotonic solution (1% Sörensen phosphate buffer (v/v), 37°C). When the surgical procedure was finished, the abdomen was covered with a cotton wool pad avoiding peritoneal liquid evaporation and heat losses. The drug solution was introduced inside the compartment and the samples were collected every 5 min up to a period of 30 min. In situ experiments were carried out with the three formulations (Co-Diovan, VALS_BE/HCTZ_Low and VALS_Supra/HCTZ_BE) predissolved in a phosphate buffer solution at pH 7 [13].

At the end of the experiments the animals were euthanized. In order to separate solid components from the samples, they were centrifuged 5 minutes at 5000 r.p.m. All samples were analyzed for VALS concentration by High Performance Liquid Chromatography (HPLC) with a previously validated procedure with adequate precision and accuracy and covering the range of the experimental samples.

At the end of the experiments there is a reduction in the volume of the perfused solutions due to water reabsorption, consequently, a correction became necessary to calculate the absorption rate constants accurately. Water reabsorption was characterized as an apparent zero order process. A method based on direct measurement of the remaining volume of the test solution was employed to calculate the water reabsorption zero order constant (ko). The volume at the beginning of the experiment (V0) is composed from the volume of the drug solution (4 mL for jejunum and ileum, 5 mL for colon and 10 mL for complete small intestine) plus the residual volume after flushing the intestinal segment. This residual volume was previously characterized and is on average 0.3 to 0.5 mL. The volume at the end of the experiment (Vend) was measured for each animal by carefully extracting and squeezing the intestinal segment. An individual value of ko was estimated for each animal as:

k_o=(V_0-V_end )/t_end                                                                                    Equation 1

where Vend is the measured volume at the end of the experiment (tend = 30 min) in each animal. ko value was used to estimate the remaining water volume in the different segments at each time point (Vt). Finally, the experimental analysed drug concentrations (Ce) were corrected at each time point to obtain the actual Ct by the following equation:

C_t=C_e (V_t⁄V_0 )                                                                                                  Equation 2

where Ct represents the drug gut concentration in the absence of any water reabsorption at time t, and Ce represents the actual experimental value. The Ct values (corrected concentrations) were used to calculate the actual absorption rate coefficients (Tugcu-Demiroz et al., 2014).

The absorption rate coefficient (ka) was determined by nonlinear regression analysis of the remaining concentrations in lumen (Ct) versus time.

C_t=C_0 e^(〖-k〗_a t)                                                                                                       Equation 3         

This ka value was transformed into permeability value with the following relationship:

P_eff=k_a R/2                                                                                                     Equation 4

where R is the effective radius of the intestinal segment. R was calculated considering the intestinal segment as a cylinder with the relationship:

Volume=πR^2 L                                                                                                Equation 5

Estimation was done using a 10 mL perfusion volume for complete small intestine. The intestinal length (L) was 100 cm for complete small intestine.

  1. Why hydrochlorothiazide was not quantified and modeled? Lines 345-346 – potential reasons for lack of hydrochlorothiazide bioequivalence – is speculation, that does not match the results in Fig. 3 (differences between the formulations in the different chambers). Please consider to analyze the hydrochlorothiazide data and to present the results of this analysis.

Figure 3 shows Valsartan amounts not HCTZ amounts. HCTZ was not quantified in the GIS experiments as the dissolution results in USP II apparatus showed no differences in its dissolution (data not shown) and the hypothesis to explain its failure was a change in a physiological variable that would have required to perform dissolution experiments in GIS at three different gastric emptying times and our main purpose was to explain VALS behavior. 

Small corrections:

  1. The terms in the title are non-standard and also appear to be misleading, especially given the discrepancies between the observed and predicted data (see above). The data actually originated from in vitro model and in vivo experiments (in situ perfusion and PK data from BEQ studies), and they were analyzed using a sort of minimal PBPK. Please revise and correct.

The discrepancies in the in vivo predictions are minimal an the key PK parameters are predicted within the stablished error limits

In vivo predictive dissolution (iPD) is not a new or non-standard term and physiologically-based biopharmaceutic in silico model reflect the nature of the developed model but as the reviewer consider is a minimal PBPK model we will change the title to:

In vivo predictive dissolution (iPD) and biopharmaceutic-based in silico model to explain bioequivalence results of valsartan, a BCS Class IV drug

  1. Line 17 – “hydrochlorothiazide” – please add the missing h in thiazide. DONE
  2. Line 136 – Opadry type and color – please check for typo errors. DONE
  3. Line 282 – “radius”, please correct. DONE

Reviewer 3 Report

Comments and Suggestions for Authors

Nice job. I've got two questions:

1. What software environment did you use for modeling?

2. Which method did you use for convolution? Was it numerical or reverse Wagner-Nelson?

Author Response

Thanks for your comments

  1. What software environment did you use for modeling?

Model development and fitting was done in Phoenix WinNonlin V8 (Certara USA, Princeton, NJ, USA).

This information has been included in the manuscript

  1. Which method did you use for convolution? Was it numerical or reverse Wagner-Nelson?

It was numerical convolution by integration of the differential equations

This information has been included in the manuscript

Round 2

Reviewer 2 Report

Comments and Suggestions for Authors

The authors made the requested changes and corrections, and explained the reasons for certain assumptions used in data analysis.

This is an interesting paper, and I recommend to approve the publication of the revised manuscript.

Please re-check the format of the citations in the text of the manuscript (Introduction, Methods, ...). This is the only minor correction that may be necessary.

Reviewer 3 Report

Comments and Suggestions for Authors

Thank you! It is a go